# On Weighted $(k, s)$-Riemann-Liouville Fractional Operators and Solution of Fractional Kinetic Equation

**Muhammad Samraiz** [1], **Muhammad Umer** [1], **Artion Kashuri** [2], **Thabet Abdeljawad** [3,4,5,*], **Sajid Iqbal** [6] **and Nabil Mlaiki** [3]

1   Department of Mathematics, University of Sargodha, Sargodha 40100, Pakistan;
    muhammad.samraiz@uos.edu.pk (M.S.); msamraizuos@gmail.com or mianumerlink4u99@gmail.com (M.U.)
2   Department of Mathematics, Faculty of Technical Science, University Ismail Qemali, 9400 Vlora, Albania;
    artionkashuri@gmail.com or artion.kashuri@univlora.edu.al
3   Department of Mathematics and General Sciences, Prince Sultan University, Riyadh 12345, Saudi Arabia;
    nmlaiki@psu.edu.sa
4   Department of Medical Research, China Medical University, Taichung 40402, Taiwan
5   Department of Computer Science and Information Engineering, Asia University, Taichung 40402, Taiwan
6   Department of Mathematics, Riphah International University, Faisalabad Campus, Satyana Road,
    Faisalabad 38000, Pakistan; sajid_uos2000@yahoo.com
*   Correspondence: tabdeljawad@psu.edu.sa

**Abstract:** In this article, we establish the weighted $(k, s)$-Riemann-Liouville fractional integral and differential operators. Some certain properties of the operators and the weighted generalized Laplace transform of the new operators are part of the paper. The article consists of Chebyshev-type inequalities involving a weighted fractional integral. We propose an integro-differential kinetic equation using the novel fractional operators and find its solution by applying weighted generalized Laplace transforms.

**Keywords:** weighted $(k, s)$ fractional integral operator; weighted $(k, s)$ fractional derivative; weighted generalized Laplace transform; fractional kinetic equation

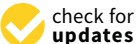



## 1. Introduction

Fractional calculus history dates back to the 17th century, when the derivative of order $\alpha = 1/2$ was defined by Leibnitz in 1695. Fractional calculus has gained broad significance in the last few decades due to its applications in various fields of science and engineering. The Tautocrone problem can be solved using fractional calculus, as shown by Abel [1]. It also has applications in group theory, field theory, polymers, continuum mechanics, wave theory, quantum mechanics, biophysics, spectroscopy, Lie theory, and in several other fields [2–6]. Despite the fact that this calculus is ancient, it has gained attention over the last few decades because of the interesting results derived when this calculus is applied to the models of some real-world problems [7–14]. The fact that there are various fractional operators is what makes fractional calculus special. Thus, any scientist working on modeling real global phenomena can choose the operator that best suits the model.

The Riemann-Liouville, Grünwald-Letnikov, and Caputo and Hadamard definitions [7,15,16] are some of the most well-known definitions of fractional operators, such that their formulations include single-kernel integrals, and they are used to explore and analyze memory effect problems, for example [17]. The fractional derivatives are represented by the fractional integrals [7,10,15,18] in fractional calculus. There are several varieties of fractional integrals, of which two have been studied extensively for their applications. The first one is the Riemann-Liouville fractional integral defined for parameter $\beta \in \mathbb{R}^+$ by

$$(\mathfrak{I}_{a^+}^{\beta} f)(\xi) = \frac{1}{\Gamma(\beta)} \int_a^{\xi} (\xi - t)^{\beta - 1} \varphi(t) dt, \ \ \beta > 0, \ \xi > a,$$

inspired by Cauchy's integral formula

$$\int_a^\xi dt_1 \int_a^{t_1} dt_2 \cdots \int_a^{t_{n-1}} dt_n = \frac{1}{\Gamma(n)} \int_a^\xi (\xi - t)^{n-1} \varphi(t) dt,$$

well-defined for $n \in N$. The second is Hadamard's fractional integral, which is defined by Hadamard [19]

$$(\mathfrak{J}_a^\beta \varphi)(\xi) = \frac{1}{\Gamma(\beta)} \int_a^\xi (log \frac{\xi}{t})^{\beta-1} \frac{\varphi(t)}{t} dt, \quad \beta > 0, \xi > a,$$

and is derived by the following integral:

$$\int_a^\xi \frac{dt_1}{t_1} \int_a^{t_1} \frac{dt_2}{t_2} \cdots \int_a^{t_{n-1}} \frac{dt_n}{t_n} = \frac{1}{\Gamma(n)} \int_a^\xi (log \frac{\xi}{t})^{n-1} \frac{\varphi(t)}{t} dt.$$

We start by recalling some related results and notions.

**Definition 1** ([20])**.** *The integral form of the k-gamma function is defined by*

$$\Gamma_k(\alpha) = \int_0^\infty \xi^{\alpha-1} e^{\frac{-\xi^k}{k}} d\xi, \Re(\alpha) > 0.$$

*Clearly,* $\Gamma(\alpha) = \lim_{k \to 1} \Gamma_k(\alpha)$ *and* $\Gamma_k(\alpha) = k^{\frac{\alpha}{k}-1} \Gamma(\frac{\alpha}{k})$.

**Definition 2.** *Let* $\Re(\alpha)$, $\Re(\beta) > 0$ *and* $k > 0$, *where we have the following k-beta function*

$$B_k(\alpha, \beta) = \frac{1}{k} \int_0^1 \tau^{\frac{\alpha}{k}-1} (1-\tau)^{\frac{\beta}{k}-1} d\tau.$$

*Note that the relation between* $\Gamma_k$ *and* $B_k$ *functions is given by* $B_k(\alpha, \beta) = \frac{\Gamma_k(\alpha)\Gamma_k(\beta)}{\Gamma_k(\alpha+\beta)}$.

The $(k, s)$-Riemann-Liouville fractional integral (RLFI) [21] is given in the following definition.

**Definition 3.** *Suppose* $\varphi \in C[a, b]$, *then* $(k, s)$*-RLFI of order* $\alpha$ *is defined by*

$$({}_k^s \mathfrak{J}_{a^+}^\alpha \varphi)(\xi) = \frac{(s+1)^{1-\frac{\alpha}{k}}}{k\Gamma_k(\alpha)} \int_a^\xi (\xi^{s+1} - t^{s+1})^{\frac{\alpha}{k}-1} t^s \varphi(t) dt, \quad \xi \in [a, b], \tag{1}$$

*where* $\alpha, k > 0$ *and* $s \in \mathbb{R} \backslash \{-1\}$.

**Definition 4** ([22])**.** *Suppose* $\varphi$ *is a continuous function on* $[0, \infty)$ *and* $s, \alpha \in \mathbb{R}^+$. *Then for all* $0 < t < \xi < \infty$

$$({}_k^s \mathfrak{D}_{a^+}^\alpha \varphi)(\xi) = \frac{(s)^{\frac{\alpha-nk+k}{k}}}{k\Gamma_k(nk-\alpha)} (\xi^{1-s} \frac{d}{d\xi})^n \int_a^\xi (\xi^s - t^s)^{\frac{nk-\alpha}{k}-1} t^{s-1} \varphi(t) dt, \tag{2}$$

*where* $n = [\alpha] + 1$ *and* $k > 0$, *is called a weighted* $(k, s)$*-Riemann Liouville fractional derivative, provided it exists.*

**Definition 5** ([23]). *Let $\varphi$, $\psi \in [a, \infty)$ be a real valued function such that $\psi(\xi)$ is continuous and $\psi'(\xi) > 0$ on $[a, \infty)$. The generalized weighted Laplace transform of $\varphi$ with weight function $\omega$ defined on $[a, \infty)$ is given by*

$$L_\psi^\omega \{\varphi(x)\}(u) = \int_a^\infty e^{-u(\psi(x) - \psi(a))} \omega(x) \varphi(x) \psi'(x) dx, \tag{3}$$

*holds for all values of u.*

**Theorem 1** ([23]). *The generalized weighted Laplace transform of $\mathfrak{D}_\omega^n \varphi$ exists and is given by*

$$\mathfrak{L}_\Psi^\omega \{\mathfrak{D}_\omega^n \Phi\}(u) = u^n \mathfrak{L}_\Psi^\omega \{\Phi(\xi)\}(u) - \sum_{k=0}^{n-1} u^{n-k-1} \Phi_k(a).$$

**Definition 6** ([23]). *The generalized weighted convolution of $\varphi$ and $\psi$ is defined by*

$$(\Phi *_\Psi^\omega h)(\xi) = \omega^{-1}(\xi) \int_a^\xi \omega(\Psi^{-1}(\Psi(\xi) + \Psi(a) - \Psi(t)))$$
$$\times \Phi(\Psi^{-1}(\Psi(\xi) + \Psi(a) - \Psi(t))) \omega(t) h(t) \Psi'(t) dt.$$

## 2. Weighted $(k, s)$-Riemann Liouville Fractional Operators

In the present section, we define the weighted $(k, s)$-Riemann Liouville fractional operators and discuss some of their properties.

**Definition 7.** *Let $\varphi$ be a continuous function on $[a, b]$. Then, the weighted $(k, s)$-RLFI of order $\alpha$ is defined by*

$$({}_k^s \mathfrak{I}_{a+, \omega}^\alpha \varphi)(\xi) = \frac{(s+1)^{1 - \frac{\alpha}{k}} \omega^{-1}(\xi)}{k \Gamma_k(\alpha)} \int_a^\xi (\xi^{s+1} - t^{s+1})^{\frac{\alpha}{k} - 1} t^s \omega(t) \varphi(t) dt, \ \xi \in [a, b], \tag{4}$$

*where $\alpha, k > 0$, $\omega(\xi) \neq 0$ and $s \in \mathbb{R} \backslash \{-1\}$.*

**Remark 1.** *It should be noted that this integral operator covers many fractional integral operators.*

(i)   *If we choose $\omega(\xi) = 1$, we obtain $(k, s)$-RLFI [21].*
(ii)  *If we choose $s = 0$ and $\omega(\xi) = 1$, k-RLFI is obtained [24].*
(iii) *For $k = 1$, $s = 0$ and $\omega(\xi) = 1$, it gives RLFI [7].*
(iv)  *For $s \to -1^+$ and $\omega(\xi) = 1$, it is converted to the k-Hadamard fractional integral [25].*

The following modification of Definition 4 is required to prove the claimed results.

**Definition 8.** *The $(k, s)$-Riemann Liouville fractional derivative is defined as follows:*
*Let $\varphi$ be a continuous function on $[0, \infty)$ and $s \in \mathbb{R} \backslash \{-1\}$. Then for all $0 < t < \xi < \infty$*

$$({}_k^s \mathfrak{D}_{a+}^\alpha \varphi)(\xi) = \frac{k^{n-1}(s+1)^{\frac{\alpha - nk + k}{k}}}{\Gamma_k(nk - \alpha)} (\xi^{-s} \frac{d}{d\xi})^n \int_a^\xi (\xi^{s+1} - t^{s+1})^{\frac{nk - \alpha}{k} - 1} t^s \varphi(t) dt,$$

*where $n = [\alpha] + 1$ and $\alpha, k > 0$, is called the $(k, s)$-Riemann Liouville fractional derivative, provided it exists.*

**Definition 9.** *Let $\varphi$ be a continuous function on $[0, \infty)$, $s \in \mathbb{R}\backslash\{-1\}$, $n = [\alpha] + 1$, $\alpha, k > 0$ and $\omega(\xi) \neq 0$. Then for all $0 < t < \xi < \infty$*

$$({}^{s}_{k}\mathfrak{D}^{\alpha}_{a+,\omega}\varphi)(\xi) = \omega^{-1}(\xi)(k\xi^{-s}\frac{d}{d\xi})^{n}\omega(\xi)({}^{s}_{k}\mathfrak{J}^{nk-\alpha}_{a+,\omega}\varphi)(t), \tag{5}$$

*where ${}^{s}_{k}\mathfrak{J}^{nk-\alpha}_{a+,\omega}$ is a weighted $(k, s)$-RLFI.*

It can also be written as

$$({}^{s}_{k}\mathfrak{D}^{\alpha}_{a+,\omega}\varphi)(\xi) = \frac{k^{n-1}(s+1)^{\frac{\alpha-nk+k}{k}}\omega^{-1}(\xi)}{\Gamma_{k}(nk-\alpha)}(\xi^{-s}\frac{d}{d\xi})^{n}$$

$$\times \int_{a}^{\xi}(\xi^{s+1} - t^{s+1})^{\frac{nk-\alpha}{k}-1}t^{s}\omega(t)\varphi(t)dt. \tag{6}$$

**Remark 2.** *It is worth mentioning that many other derivative operators can be represented as special cases of (6).*

(i) *If $\omega(\xi) = 1$ is chosen, we obtain the $(k, s)$-Riemann-Liouville fractional derivative [22].*
(ii) *Let $s = 0$ and $\omega(\xi) = 1$, where it gives the $k$-Riemann-Liouville fractional derivative [26].*
(iii) *For $k = 1$, $s = 0$ and $\omega(\xi) = 1$, it reduces to the Riemann-Liouville fractional derivative [27].*
(iv) *It reduces to the $k$-Hadamard fractional derivative for $s \to -1^{+}$, $\omega(\xi) = 1$ [25].*

Next, we present the space where the weighted $(k, s)$-Riemann-Liouville fractional integrals are bounded.

**Definition 10.** *Let $\varphi$ be a function defined on $[a, b]$. The space $X^{p}_{\omega}(a, b)$, $1 \leq p \leq \infty$ is the space of all Lebesgue measurable functions for which $\| \varphi \|_{X^{p}_{\omega}} < \infty$, where*

$$\| \varphi \|_{X^{p}_{\omega}} = \left[(s+1)\int_{a}^{b} | \omega(\xi)\varphi(\xi) |^{p} \xi^{s}d\xi\right]^{\frac{1}{p}}, \;\; 1 \leq p < \infty,$$

*$\omega(\xi) \neq 0$, $s \in \mathbb{R}$ and*

$$\| \varphi \|_{X^{\infty}_{\omega}} = ess\,sup_{a \leq \xi \leq b} | \omega(\xi)\varphi(\xi) | < \infty.$$

Noted that $\varphi \in X^{p}_{\omega}(a, b) \Leftrightarrow \omega(\xi)\varphi(\xi)(\xi^{s})^{\frac{1}{p}} \in L_{p}(a, b)$ for $1 \leq p < \infty$ and $\varphi \in X^{\infty}_{\omega}(a, b)$ $\Leftrightarrow \omega(\xi)\varphi(\xi) \in L_{\infty}(a, b)$.

**Theorem 2.** *Let $\alpha > 0$, $k > 0$, $1 \leq p \leq \infty$ and $\varphi \in X^{p}_{\omega}(a, b)$. Then ${}^{s}_{k}\mathfrak{J}^{\alpha}_{a+,\omega}\varphi$ is bounded in $X^{p}_{\omega}(a, b)$ and*

$$\|{}^{s}_{k}\mathfrak{J}^{\alpha}_{a+,\omega}\varphi \|_{X^{p}_{\omega}} \leq \frac{(s+1)^{-\frac{\alpha}{k}}(b^{s+1} - a^{s+1})^{\frac{\alpha}{k}}}{\Gamma_{k}(\alpha+1)} \| \varphi \|_{X^{p}_{\omega}}.$$

**Proof.** For $1 \leq p < \infty$, we have

$$\|{}^{s}_{k}\mathfrak{J}^{\alpha}_{a+,\omega}\varphi \|_{X^{p}_{\omega}} = \left[(s+1)\int_{a}^{b}\left|\omega(\xi)\frac{(s+1)^{1-\frac{\alpha}{k}}\omega^{-1}(\xi)}{k\Gamma_{k}(\alpha)}\right.\right.$$

$$\left.\left.\times \int_{a}^{\xi}(\xi^{s+1} - t^{s+1})^{\frac{\alpha}{k}-1}t^{s}\omega(t)\varphi(t)dt\right|^{p}\xi^{s}d\xi\right]^{\frac{1}{p}}$$

$$= \frac{(s+1)^{2-\frac{\alpha}{k}}}{k\Gamma_{k}(\alpha)}\left[\int_{a}^{b}\left|\int_{a}^{\xi}(\xi^{s+1} - t^{s+1})^{\frac{\alpha}{k}-1}t^{s}\omega(t)\varphi(t)dt\right|^{p}\xi^{s}d\xi\right]^{\frac{1}{p}}. \tag{7}$$

Substituting $\xi^{s+1} = v$ and $t^{s+1} = u$ on the right side of (7), we obtain

$$\| {}_k^s \mathfrak{I}_{a^+,\omega}^\alpha \varphi \|_{X_\omega^p} = \frac{(s+1)^{2-\frac{\alpha}{k}}}{k\Gamma_k(\alpha)} \Big[ \int_{a^{s+1}}^{b^{s+1}} \Big| \int_{a^{s+1}}^v (v-u)^{\frac{\alpha}{k}-1} \omega(u^{\frac{1}{s+1}}) \varphi(u^{\frac{1}{s+1}}) du \Big|^p dv \Big]^{\frac{1}{p}}.$$

By using Minkowski's inequality, we have

$$\| {}_k^s \mathfrak{I}_{a^+,\omega}^\alpha \varphi \|_{X_\omega^p} \le \frac{(s+1)^{-\frac{\alpha}{k}}}{k\Gamma_k(\alpha)} \int_{a^{s+1}}^{b^{s+1}} \Big[ \int_u^{a^{s+1}} \Big| (v-u)^{\frac{\alpha}{k}-1} \omega(u^{\frac{1}{s+1}}) \varphi(u^{\frac{1}{s+1}}) dv \Big|^p du \Big]^{\frac{1}{p}}$$

$$\le \frac{(s+1)^{-\frac{\alpha}{k}}}{k\Gamma_k(\alpha)} \int_{a^{s+1}}^{b^{s+1}} \Big| \omega(u^{\frac{1}{s+1}}) \varphi(u^{\frac{1}{s+1}}) \Big| \Big[ \frac{(b^{s+1}-u)^{(\frac{\alpha}{k}-1)p+1}}{(\frac{\alpha}{k}-1)p+1} \Big]^{\frac{1}{p}} du.$$

Applying Hölder's inequality, we obtain

$$\| {}_k^s \mathfrak{I}_{a^+,\omega}^\alpha \varphi \|_{X_\omega^p} \le \frac{(s+1)^{-\frac{\alpha}{k}}}{k\Gamma_k(\alpha)} \Big[ \int_{a^{s+1}}^{b^{s+1}} \Big| \omega(u^{\frac{1}{s+1}}) \varphi(u^{\frac{1}{s+1}}) \Big|^p du \Big]^{\frac{1}{p}}$$

$$\times \Big[ \int_{a^{s+1}}^{b^{s+1}} \Big( \frac{(b^{s+1}-u)^{(\frac{\alpha}{k}-1)p+1}}{(\frac{\alpha}{k}-1)p+1} \Big)^{\frac{q}{p}} du \Big]^{\frac{1}{q}},$$

where $\frac{1}{p} + \frac{1}{q} = 1$. Further,

$$\| {}_k^s \mathfrak{I}_{a^+,\omega}^\alpha \varphi \|_{X_\omega^p} \le \frac{(s+1)^{-\frac{\alpha}{k}}}{k\Gamma_k(\alpha)} \Big[ \int_a^b \Big| \omega(t)\varphi(t) \Big|^p (s+1) dt \Big]^{\frac{1}{p}}$$

$$\times \Big[ \int_{a^{s+1}}^{b^{s+1}} \Big( \frac{(b^{s+1}-u)^{(\frac{\alpha}{k}-1)p+1}}{(\frac{\alpha}{k}-1)p+1} \Big)^{\frac{q}{p}} du \Big]^{\frac{1}{q}}$$

$$\le \frac{(s+1)^{-\frac{\alpha}{k}} (b^{s+1}-a^{s+1})^{\frac{\alpha}{k}}}{k\Gamma_k(\alpha)\frac{\alpha}{k}} \| \varphi \|_{X_\omega^p}$$

$$= \frac{(s+1)^{-\frac{\alpha}{k}} (b^{s+1}-a^{s+1})^{\frac{\alpha}{k}}}{\Gamma_k(\alpha+1)} \| \varphi \|_{X_\omega^p}.$$

For $p = \infty$, we obtain

$$\Big| \omega(\xi) {}_k^s \mathfrak{I}_{a^+,\omega}^\alpha \varphi(\xi) \Big| = \frac{(s+1)^{-\frac{\alpha}{k}} (b^{s+1}-a^{s+1})^{\frac{\alpha}{k}}}{\Gamma_k(\alpha+1)} \| \varphi \|_{X_\omega^\infty}.$$

Hence, we obtain the desired result. $\square$

**Theorem 3.** *Let $\varphi$ be a continuous function on $[0,\infty)$ and $s \in \mathbb{R}\backslash\{-1\}$ and $\omega(\xi) \ne 0$, $n = [\alpha]+1$. Then for all $0 < a < \xi$, we obtain*

$$ {}_k^s \mathfrak{D}_{a,\omega}^\alpha ( {}_k^s \mathfrak{I}_{a^+,\omega}^\alpha \varphi)(\xi) = \varphi(\xi),$$

*where $\alpha, k > 0$.*

**Proof.** Consider

$$
{}_k^s\mathfrak{D}^\alpha_{a^+,\omega}({}_k^s\mathfrak{J}^\alpha_{a^+,\omega}\varphi)(\xi)
$$

$$
= \frac{(s+1)^{\frac{\alpha-nk+k}{k}}\omega^{-1}(\xi)}{k\Gamma_k(nk-\alpha)}(\xi^{-s}\frac{d}{d\xi})^n k^n
$$

$$
\times \int_a^\xi (\xi^{s+1}-y^{s+1})^{\frac{nk-\alpha}{k}-1}y^s\omega(y)({}_k^s\mathfrak{J}^\alpha_{a^+,\omega}\varphi)(y)dy
$$

$$
= \frac{(s+1)^{\frac{\alpha-nk+k}{k}}\omega^{-1}(\xi)}{k\Gamma_k(nk-\alpha)}(\xi^{-s}\frac{d}{d\xi})^n k^n \int_a^\xi (\xi^{s+1}-y^{s+1})^{\frac{nk-\alpha}{k}-1}y^s\omega(y)
$$

$$
\times \frac{(s+1)^{\frac{1-\alpha}{k}}\omega^{-1}(\xi)}{k\Gamma_k(\alpha)} \int_a^y (y^{s+1}-t^{s+1})^{\frac{\alpha}{k}-1}t^s\omega(t)(t)dt
$$

$$
= \frac{(s+1)^{2-n}\omega^{-1}(\xi)}{k^2\Gamma_k(\alpha)\Gamma_k(nk-\alpha)}(\xi^{-s}\frac{d}{d\xi})^n k^n
$$

$$
\times \int_a^\xi t^s\omega(t)\varphi(t)\Big[\int_t^\xi (y^{s+1}-t^{s+1})^{\frac{\alpha}{k}-1}(\xi^{s+1}-y^{s+1})^{\frac{nk-\alpha}{k}-1}y^s dy\Big]dt. \tag{8}
$$

By substituting $z = \frac{y^{s+1}-t^{s+1}}{\xi^{s+1}-t^{s+1}}$ on the right side of (8), we obtain

$$
{}_k^s\mathfrak{D}^\alpha_{a^+,\omega}({}_k^s\mathfrak{J}^\alpha_{a^+,\omega}\varphi)(\xi)
$$

$$
= \frac{(s+1)^{1-n}\omega^{-1}(\xi)}{k^2\Gamma_k(\alpha)\Gamma_k(nk-\alpha)}(\xi^{-s}\frac{d}{d\xi})^n k^n
$$

$$
\times \int_a^\xi t^s\omega(t)\varphi(t)(\xi^{s+1}-t^{s+1})^{n-1}\Big[\int_t^\xi (1-z)^{\frac{\alpha}{k}-1}(z)^{\frac{nk-\alpha}{k}-1}dz\Big]dt
$$

$$
= \frac{(s+1)^{1-n}\omega^{-1}(\xi)}{k^2\Gamma_k(\alpha)\Gamma_k(nk-\alpha)}(\xi^{-s}\frac{d}{d\xi})^n k^n
$$

$$
\times \int_a^\xi t^s\omega(t)\varphi(t)(\xi^{s+1}-t^{s+1})^{n-1}[kB_k(\alpha,nk-\alpha)]dt
$$

$$
= \frac{(s+1)^{1-n}\omega^{-1}(\xi)}{k\Gamma_k(nk)}(\xi^{-s}\frac{d}{d\xi})^n k^n \int_a^\xi t^s\omega(t)\varphi(t)(\xi^{s+1}-t^{s+1})^{n-1}dt
$$

$$
= \frac{(s+1)^{1-n}\omega^{-1}(\xi)}{k^n\Gamma(n)}(\xi^{-s}\frac{d}{d\xi})^n k^n \int_a^\xi t^s\omega(t)\varphi(t)(\xi^{s+1}-t^{s+1})^{n-1}dt,
$$

which gives

$$
{}_k^s\mathfrak{D}^\alpha_{a^+,\omega}({}_k^s\mathfrak{J}^\alpha_{a^+,\omega}\varphi)(\xi) = \varphi(\xi).
$$

The inverse property is proved. $\square$

**Corollary 1.** *Let $\varphi$ be a continuous function on $[0,\infty)$ and $s \in \mathbb{R}\backslash\{-1\}$ and $\omega(\xi) \neq 0$, $m = [\beta] + 1$, $n = [\alpha] + 1$. Then for all $0 < a < \xi$*

$$
{}_k^s\mathfrak{D}^\alpha_{a^+,\omega}({}_k^s\mathfrak{J}^\beta_{a^+,\omega}\varphi)(\xi) = ({}_k^s\mathfrak{D}^{\alpha-\beta}_{a^+,\omega}\varphi)(\xi),
$$

*where $\alpha, \beta, k > 0$.*

**Corollary 2.** *(Semi-group property) Let $\varphi$ be a continuous function on $[0,\infty)$ and $s \in \mathbb{R}\backslash\{-1\}$, $\omega(\xi) \neq 0$, $n = [\alpha] + 1$, $m = [\beta] + 1$ and $\alpha + \beta < nk$. Then for all $0 < a < \xi$*

$$
{}_k^s\mathfrak{D}^\alpha_{a^+,\omega}({}_k^s\mathfrak{D}^\beta_{a^+,\omega}\varphi)(\xi) = ({}_k^s\mathfrak{D}^{\alpha+\beta}_{a^+,\omega}\varphi)(\xi),
$$

*where $\alpha, \beta, k > 0$.*

**Proof.** By using Definition 9, we have

$$
\begin{aligned}
{}^{s}_{k}\mathfrak{D}^{\alpha}_{a^{+},\omega}({}^{s}_{k}\mathfrak{D}^{\beta}_{a^{+},\omega}\varphi)(\xi)
&= \omega^{-1}(\xi)(k\xi^{-s}\frac{d}{d\xi})^{n}\omega(\xi)({}^{s}_{k}\mathfrak{J}^{nk-\alpha}_{a^{+},\omega})({}^{s}_{k}\mathfrak{D}^{\beta}_{a^{+},\omega}\varphi)(\xi) \\
&= \omega^{-1}(\xi)(k\xi^{-s}\frac{d}{d\xi})^{n}\omega(\xi)({}^{s}_{k}\mathfrak{J}^{nk-\alpha}_{a^{+},\omega})({}^{s}_{k}\mathfrak{D}^{\beta}_{a^{+},\omega}\varphi)(\xi)({}^{s}_{k}\mathfrak{J}^{\beta}_{a^{+},\omega})({}^{s}_{k}\mathfrak{J}^{-\beta}_{a^{+},\omega}).
\end{aligned}
$$

By using Theorem 3, we have

$$
\begin{aligned}
{}^{s}_{k}\mathfrak{D}^{\alpha}_{a^{+},\omega}({}^{s}_{k}\mathfrak{D}^{\beta}_{a^{+},\omega}\varphi)(\xi)
&= \omega^{-1}(\xi)(k\xi^{-s}\frac{d}{d\xi})^{n}\omega(\xi)({}^{s}_{k}\mathfrak{J}^{nk-\alpha}_{a^{+},\omega})({}^{s}_{k}\mathfrak{J}^{-\beta}_{a^{+},\omega}) \\
&= \omega^{-1}(\xi)(k\xi^{-s}\frac{d}{d\xi})^{n}\omega(\xi)({}^{s}_{k}\mathfrak{J}^{nk-(\alpha+\beta)}_{a^{+},\omega}),
\end{aligned}
$$

which implies

$$
{}^{s}_{k}\mathfrak{D}^{\alpha}_{a^{+},\omega}({}^{s}_{k}\mathfrak{D}^{\beta}_{a^{+},\omega}\varphi)(\xi) = ({}^{s}_{k}\mathfrak{D}^{\alpha+\beta}_{a^{+},\omega}\varphi)(\xi),
$$

which is the required result. $\square$

**Corollary 3** (Commutative property). *Let $\varphi$ be a continuous function on $[0,\infty)$ and $\alpha,\beta \in \mathbb{R}^{+}$, $\omega(\xi) \neq 0$ and $s \in \mathbb{R}\backslash\{-1\}$. Then for all $0 < a < \xi$*

$$
{}^{s}_{k}\mathfrak{D}^{\alpha}_{a^{+},\omega}({}^{s}_{k}\mathfrak{D}^{\beta}_{a^{+},\omega}\varphi)(\xi) = {}^{s}_{k}\mathfrak{D}^{\beta}_{a^{+},\omega}({}^{s}_{k}\mathfrak{D}^{\alpha}_{a^{+},\omega}\varphi)(\xi).
$$

**Corollary 4** (Linearity property). *Let $\varphi$ be a continuous function on $[0,\infty)$, $k,\alpha \in \mathbb{R}^{+}$, $\omega(\xi) \neq 0$ and $s \in \mathbb{R}\backslash\{-1\}$. Then for all $0 < a < \xi$*

$$
{}^{s}_{k}\mathfrak{D}^{\alpha}_{a^{+},\omega}[\psi(\xi) + \mu h(\xi)] = {}^{s}_{k}\mathfrak{D}^{\alpha}_{a^{+},\omega}\psi(\xi) + \mu {}^{s}_{k}\mathfrak{D}^{\alpha}_{a^{+},\omega}h(\xi),
$$

*where $n \in N$ and $n = [\alpha] + 1$.*

**Theorem 4.** *Let $\varphi$ be a continuous function on $[a,b]$, $k > 0$, $\omega(\xi) \neq 0$ and $s \in \mathbb{R}\backslash\{-1\}$*

$$
{}^{s}_{k}\mathfrak{J}^{\beta}_{a^{+},\omega}[{}^{s}_{k}\mathfrak{J}^{\alpha}_{a^{+},\omega}\varphi(\xi)] = {}^{s}_{k}\mathfrak{J}^{\alpha}_{a^{+},\omega}[{}^{s}_{k}\mathfrak{J}^{\beta}_{a^{+},\omega}\varphi(\xi)] = {}^{s}_{k}\mathfrak{J}^{\alpha+\beta}_{a^{+},\omega}\varphi(\xi),
$$

*for all $\alpha,\ \beta > 0$ and $\xi \in [a,b]$.*

**Proof.** By using Definition 7 and Dirichlet's formula, we obtain

$$
\begin{aligned}
&{}^{s}_{k}\mathfrak{J}^{\alpha}_{a^{+},\omega}[{}^{s}_{k}\mathfrak{J}^{\beta}_{a^{+},\omega}\varphi(\xi)] \\
&= \frac{(s+1)^{1-\frac{\alpha}{k}}\omega^{-1}(\xi)}{k\Gamma_{k}(\alpha)}\int_{a}^{\xi}(\xi^{s+1}-t^{s+1})^{\frac{\alpha}{k}-1}t^{s}\omega(t){}^{s}_{k}\mathfrak{J}^{\beta}_{a^{+},\omega}\varphi(t)dt \\
&= \frac{(s+1)^{1-\frac{\alpha}{k}}\omega^{-1}(\xi)}{k\Gamma_{k}(\alpha)}\int_{a}^{\xi}(\xi^{s+1}-t^{s+1})^{\frac{\alpha}{k}-1}t^{s}\omega(t)\varphi(\tau) \\
&\quad \times \left[\frac{(s+1)^{1-\frac{\beta}{k}}\omega^{-1}(t)}{k\Gamma_{k}(\beta)}\int_{a}^{t}(t^{s+1}-\tau^{s+1})^{\frac{\beta}{k}-1}\tau^{s}\omega(\tau)d\tau\right]dt \\
&= \frac{(s+1)^{2-\frac{\alpha+\beta}{k}}\omega^{-1}(\xi)}{k^{2}\Gamma_{k}(\alpha)\Gamma_{k}(\beta)}\int_{a}^{\xi}\tau^{s}\omega(\tau)\varphi(\tau) \\
&\quad \times \int_{\tau}^{\xi}(\xi^{s+1}-t^{s+1})^{\frac{\alpha}{k}-1}(t^{s+1}-\tau^{s+1})^{\frac{\beta}{k}-1}t^{s}dtd\tau.
\end{aligned}
\tag{9}
$$

By substituting $y = \frac{t^{s+1} - \tau^{s+1}}{\xi^{s+1} - \tau^{s+1}}$ on the right side of (9), we obtain

$$
\begin{aligned}
&{}_k^s \mathfrak{J}_{a^+,\omega}^{\alpha} [{}_k^s \mathfrak{J}_{a^+,\omega}^{\beta} \varphi(\xi)] \\
&= \frac{(s+1)^{2 - \frac{\alpha+\beta}{k}} \omega^{-1}(\xi)}{k^2 \Gamma_k(\alpha) \Gamma_k(\beta)} \\
&\quad \times \int_a^{\xi} \frac{(\xi^{s+1} - \tau^{s+1})^{\frac{\alpha+\beta}{k} - 1}}{(s+1)} k B_k(\alpha, \beta) \tau^s \omega(\tau) \varphi(\tau) d\tau \\
&= \frac{(s+1)^{1 - \frac{\alpha+\beta}{k}} \omega^{-1}(\xi)}{k \Gamma_k(\alpha+\beta)} \int_a^{\xi} (\xi^{s+1} - \tau^{s+1})^{\frac{\alpha+\beta}{k} - 1} \tau^s \omega(\tau) \varphi(\tau) d\tau \\
&= {}_k^s \mathfrak{J}_{a^+,\omega}^{\alpha+\beta} \varphi(\xi).
\end{aligned}
$$

The proof is completed. □

**Theorem 5.** *Let $\alpha, \beta, k > 0$, $\omega(\xi) \neq 0$ and $s \in \mathbb{R} \backslash \{-1\}$. Then we have*

$$
{}_k^s \mathfrak{J}_{a^+,\omega}^{\beta} [\omega^{-1}(\xi)(\xi^{s+1} - a^{s+1})^{\frac{\beta}{k} - 1}] = \frac{\Gamma_k(\beta)(\xi^{s+1} - a^{s+1})^{\frac{\alpha+\beta}{k} - 1} \omega^{-1}(\xi)}{(s+1)^{\frac{\alpha}{k}} \Gamma_k(\alpha+\beta)},
$$

*where $\Gamma_k$ denotes the k-Gamma function.*

**Proof.** By using Definition 7, we obtain

$$
\begin{aligned}
&{}_k^s \mathfrak{J}_{a^+,\omega}^{\beta} [\omega^{-1}(\xi)(\xi^{s+1} - a^{s+1})^{\frac{\beta}{k} - 1}] \\
&= \frac{(s+1)^{1 - \frac{\alpha}{k}} \omega^{-1}(\xi)}{k \Gamma_k(\alpha)} \int_a^{\xi} (\xi^{s+1} - t^{s+1})^{\frac{\alpha}{k} - 1} t^s \\
&\quad \times (\xi^{s+1} - a^{s+1})^{\frac{\beta}{k} - 1} \omega^{-1}(t) \omega(t) \varphi(t) dt.
\end{aligned}
\tag{10}
$$

By substituting $y = \frac{\xi^{s+1} - t^{s+1}}{\xi^{s+1} - a^{s+1}}$ on the right side of (10), we obtain

$$
\begin{aligned}
&{}_k^s \mathfrak{J}_{a^+,\omega}^{\beta} [\omega^{-1}(\xi)(\xi^{s+1} - a^{s+1})^{\frac{\beta}{k} - 1}] \\
&= \frac{(s+1)^{\frac{-\alpha}{k}} \omega^{-1}(\xi)(\xi^{s+1} - a^{s+1})^{\frac{\alpha+\beta}{k} - 1}}{k \Gamma_k(\alpha)} \\
&\quad \times \int_0^1 (1 - y)^{\frac{\alpha}{k} - 1} (y)^{\frac{\beta}{k} - 1} dy \\
&= \frac{(s+1)^{\frac{-\alpha}{k}} (\xi^{s+1} - a^{s+1})^{\frac{\alpha+\beta}{k} - 1} \omega^{-1}(\xi)}{k \Gamma_k(\alpha)} k B_k(\alpha, \beta) \\
&= \frac{\Gamma_k(\beta)(\xi^{s+1} - a^{s+1})^{\frac{\alpha+\beta}{k} - 1} \omega^{-1}(\xi)}{(s+1)^{\frac{\alpha}{k}} \Gamma_k(\alpha+\beta)}.
\end{aligned}
$$

This completes the proof. □

**Corollary 5.** *Let $k > 0$, $\omega(\xi) \neq 0$ and $s \in \mathbb{R} \backslash \{-1\}$. Then, we have*

$$
{}_k^s \mathfrak{J}_{a^+,\omega}^{\alpha} [\omega^{-1}(\xi)(1)] = \frac{(\xi^{s+1} - a^{s+1})^{\frac{\alpha}{k} - 2} \omega^{-1}(\xi)}{(s+1)^{\frac{\alpha}{k}} \Gamma_k(\alpha+\beta)}.
\tag{11}
$$

**Remark 3.** *Taking $\omega(\xi) = 1$ in Theorem 5 and Corollary 5, we obtain results of [21].*

**Remark 4.** *If we choose $s = 0$, $k = 1$ and $\omega(\xi) = 1$ in Theorem 5 and Corollary 5, we obtain results for Riemann Liouville.*

### 3. Some New Chebyshev Inequalities Involving Weighted $(k, s)$-RLFI

Weighted $(k, s)$-RLFI formulations of Chebyshev-type inequalities are as follows:

**Theorem 6.** *Let $\varphi$ and $\psi$ be two synchronous functions on $[0, \infty)$. Then for all $t > a \geq 0$ and the weighted function $\omega(\xi) \neq 0$, the following inequalities for weighted $(k, s)$-RLFI hold:*

$$
{}_{k}^{s}\mathfrak{I}_{a^+,\omega}^{\alpha}\varphi\psi(t) \geq \frac{1}{{}_{k}^{s}\mathfrak{I}_{a^+,\omega}^{\alpha}(1)}{}_{k}^{s}\mathfrak{I}_{a^+,\omega}^{\alpha}\varphi(t){}_{k}^{s}\mathfrak{I}_{a^+,\omega}^{\alpha}\psi(t) \tag{12}
$$

*and*

$$
{}_{k}^{s}\mathfrak{I}_{a^+,\omega}^{\alpha}\varphi\psi(t){}_{k}^{s}\mathfrak{I}_{a^+,\omega}^{\beta}(1) + {}_{k}^{s}\mathfrak{I}_{a^+,\omega}^{\beta}\varphi\psi(t){}_{k}^{s}\mathfrak{I}_{a^+,\omega}^{\alpha}(1)
$$
$$
\geq \quad {}_{k}^{s}\mathfrak{I}_{a^+,\omega}^{\alpha}\varphi(t){}_{k}^{s}\mathfrak{I}_{a^+,\omega}^{\beta}\psi(t) + {}_{k}^{s}\mathfrak{I}_{a^+,\omega}^{\alpha}\psi(t){}_{k}^{s}\mathfrak{I}_{a^+,\omega}^{\beta}\varphi(t), \tag{13}
$$

*where $\alpha, \beta > 0$.*

**Proof.** Since $\varphi$ and $\psi$ are synchronous on $[0, \infty)$, for all $\xi, y \geq 0$, we have

$$
(\varphi(\xi) - \varphi(y))(\psi(\xi) - \psi(y)) \geq 0
$$
$$
\varphi(\xi)\psi(\xi) + \varphi(y)\psi(y) \geq \varphi(\xi)\psi(y) + \varphi(y)\psi(\xi). \tag{14}
$$

Both sides of (14) are multiplied by $\frac{(s+1)^{1-\frac{\alpha}{k}}\omega^{-1}(t)}{k\Gamma_k(\alpha)}(t^{s+1} - \xi^{s+1})^{\frac{\alpha}{k}-1}\omega(\xi)\xi^s$ and integrating w.r.t $\xi$ over (a,t), we obtain

$$
\frac{(s+1)^{1-\frac{\alpha}{k}}\omega^{-1}(t)}{k\Gamma_k(\alpha)}\int_a^t (t^{s+1} - \xi^{s+1})^{\frac{\alpha}{k}-1}\xi^s\omega(\xi)\varphi(\xi)\psi(\xi)d\xi
$$
$$
+ \varphi(y)\psi(y)\frac{(s+1)^{1-\frac{\alpha}{k}}\omega^{-1}(t)}{k\Gamma_k(\alpha)}\int_a^t (t^{s+1} - \xi^{s+1})^{\frac{\alpha}{k}-1}\xi^s\omega(\xi)d\xi
$$
$$
\geq \psi(y)\frac{(s+1)^{1-\frac{\alpha}{k}}\omega^{-1}(t)}{k\Gamma_k(\alpha)}\int_a^t (t^{s+1} - \xi^{s+1})^{\frac{\alpha}{k}-1}\xi^s\omega(\xi)\varphi(\xi)d\xi
$$
$$
+ \varphi(y)\frac{(s+1)^{1-\frac{\alpha}{k}}\omega^{-1}(t)}{k\Gamma_k(\alpha)}\int_a^t (t^{s+1} - \xi^{s+1})^{\frac{\alpha}{k}-1}\xi^s\omega(\xi)\psi(\xi)d\xi, \tag{15}
$$

which gives

$$
{}_{k}^{s}\mathfrak{I}_{a^+,\omega}^{\alpha}\varphi\psi(t) + \varphi(y)\psi(y){}_{k}^{s}\mathfrak{I}_{a^+,\omega}^{\alpha}(1) \geq \psi(y){}_{k}^{s}\mathfrak{I}_{a^+,\omega}^{\alpha}\varphi(t) + \varphi(y){}_{k}^{s}\mathfrak{I}_{a^+,\omega}^{\alpha}\psi(t).
$$

Both sides of (15) are multiplied by $\frac{(s+1)^{1-\frac{\alpha}{k}}\omega^{-1}(t)}{k\Gamma_k(\alpha)}(t^{s+1}-y^{s+1})^{\frac{\alpha}{k}-1}\omega(y)y^s$ and integrating w.r.t $y$ over (a,t), we obtain

$$
{}^s_k\mathfrak{J}^\alpha_{a^+,\omega}\varphi\psi(t)\frac{(s+1)^{1-\frac{\alpha}{k}}\omega^{-1}(t)}{k\Gamma_k(\alpha)}\int_a^t(t^{s+1}-y^{s+1})^{\frac{\alpha}{k}-1}y^s\omega(y)dy
$$

$$
+{}^s_k\mathfrak{J}^\alpha_{a^+,\omega}(1)\frac{(s+1)^{1-\frac{\alpha}{k}}\omega^{-1}(t)}{k\Gamma_k(\alpha)}\int_a^t(t^{s+1}-y^{s+1})^{\frac{\alpha}{k}-1}y^s\omega(y)\varphi(y)\psi(y)dy
$$

$$
\geq{}^s_k\mathfrak{J}^\alpha_{a^+,\omega}\varphi(t)\frac{(s+1)^{1-\frac{\alpha}{k}}\omega^{-1}(t)}{k\Gamma_k(\alpha)}\int_a^t(t^{s+1}-y^{s+1})^{\frac{\alpha}{k}-1}y^s\omega(y)\psi(y)dy
$$

$$
+{}^s_k\mathfrak{J}^\alpha_{a^+,\omega}\psi(t)\frac{(s+1)^{1-\frac{\alpha}{k}}\omega^{-1}(t)}{k\Gamma_k(\alpha)}\int_a^t(t^{s+1}-y^{s+1})^{\frac{\alpha}{k}-1}y^s\omega(y)\varphi(y)dy.
$$

This can be written as

$$
\begin{aligned}
&{}^s_k\mathfrak{J}^\alpha_{a^+,\omega}\varphi\psi(t){}^s_k\mathfrak{J}^\alpha_{a^+,\omega}(1)+{}^s_k\mathfrak{J}^\alpha_{a^+,\omega}\varphi\psi(t){}^s_k\mathfrak{J}^\alpha_{a^+,\omega}(1)\\
&\geq{}^s_k\mathfrak{J}^\alpha_{a^+,\omega}\varphi(t){}^s_k\mathfrak{J}^\alpha_{a^+,\omega}\psi(t)+{}^s_k\mathfrak{J}^\alpha_{a^+,\omega}\varphi(t){}^s_k\mathfrak{J}^\alpha_{a^+,\omega}\psi(t).
\end{aligned}
\tag{16}
$$

On simplification, we obtain

$$
2{}^s_k\mathfrak{J}^\alpha_{a^+,\omega}\varphi\psi(t){}^s_k\mathfrak{J}^\alpha_{a^+,\omega}(1)\geq 2{}^s_k\mathfrak{J}^\alpha_{a^+,\omega}\varphi(t){}^s_k\mathfrak{J}^\alpha_{a^+,\omega}\psi(t),
$$

which can be written as

$$
{}^s_k\mathfrak{J}^\alpha_{a^+,\omega}\varphi\psi(t)\geq\frac{1}{{}^s_k\mathfrak{J}^\alpha_{a^+,\omega}(1)}{}^s_k\mathfrak{J}^\alpha_{a^+,\omega}\varphi(t){}^s_k\mathfrak{J}^\alpha_{a^+,\omega}\psi(t).
$$

This completes the proof of (12).

Both sides of (16) are multiplied by $\frac{(s+1)^{1-\frac{\beta}{k}}\omega^{-1}(t)}{k\Gamma_k(\alpha)}(t^{s+1}-y^{s+1})^{\frac{\beta}{k}-1}\omega(y)y^s$ and integrating w.r.t $y$ over (a,t), we obtain

$$
{}^s_k\mathfrak{J}^\alpha_{a^+,\omega}\varphi\psi(t)\frac{(s+1)^{1-\frac{\beta}{k}}\omega^{-1}(t)}{k\Gamma_k(\beta)}\int_a^t(t^{s+1}-y^{s+1})^{\frac{\beta}{k}-1}y^s\omega(y)dy
$$

$$
+{}^s_k\mathfrak{J}^\alpha_{a^+,\omega}(1)\frac{(s+1)^{1-\frac{\beta}{k}}\omega^{-1}(t)}{k\Gamma_k(\beta)}\int_a^t(t^{s+1}-y^{s+1})^{\frac{\beta}{k}-1}y^s\varphi(y))\psi(y)\omega(y)dy
$$

$$
\geq\quad{}^s_k\mathfrak{J}^\alpha_{a^+,\omega}\varphi(t)\frac{(s+1)^{1-\frac{\beta}{k}}\omega^{-1}(t)}{k\Gamma_k(\beta)}\int_a^t(t^{s+1}-y^{s+1})^{\frac{\beta}{k}-1}y^s\omega(y)\psi(y)dy
$$

$$
+{}^s_k\mathfrak{J}^\alpha_{a^+,\omega}\psi(t)\frac{(s+1)^{1-\frac{\beta}{k}}\omega^{-1}(t)}{k\Gamma_k(\beta)}\int_a^t(t^{s+1}-y^{s+1})^{\frac{\beta}{k}-1}y^s\omega(y)\varphi(y)dy,
$$

which gives

$$
\begin{aligned}
&{}^s_k\mathfrak{J}^\alpha_{a^+,\omega}\varphi\psi(t){}^s_k\mathfrak{J}^\beta_{a^+,\omega}(1)+{}^s_k\mathfrak{J}^\beta_{a^+,\omega}\varphi\psi(t){}^s_k\mathfrak{J}^\alpha_{a^+,\omega}(1)\\
&\geq{}^s_k\mathfrak{J}^\alpha_{a^+,\omega}\varphi(t){}^s_k\mathfrak{J}^\beta_{a^+,\omega}\psi(t)+{}^s_k\mathfrak{J}^\alpha_{a^+,\omega}\psi(t){}^s_k\mathfrak{J}^\beta_{a^+,\omega}\varphi(t).
\end{aligned}
$$

The proof of (13) is done. $\square$

**Theorem 7.** *Let $\varphi$ and $\psi$ be two synchronous functions on $[0, \infty)$ and $h(t) \geq 0$. Then for all $t > a \geq 0$, the following inequality holds:*

$$
\frac{\omega^{-1}(\xi)}{(s+1)^{1-\frac{\beta}{k}} \Gamma_k(\beta + k)} (t^{s+1} - a^{s+1})^{\frac{\beta}{k} - 2} {}_k^s \mathfrak{J}_{a^+, \omega}^\alpha \varphi \psi h(t)
$$
$$
+ \frac{\omega^{-1}(\xi)}{(s+1)^{1-\frac{\alpha}{k}} \Gamma_k(\alpha + k)} (t^{s+1} - a^{s+1})^{\frac{\alpha}{k} - 2} {}_k^s \mathfrak{J}_{a^+, \omega}^\beta \varphi \psi h(t)
$$
$$
\geq {}_k^s \mathfrak{J}_{a^+, \omega}^\alpha \varphi h(t) {}_k^s \mathfrak{J}_{a^+, \omega}^\beta \psi(t) + {}_k^s \mathfrak{J}_{a^+, \omega}^\alpha \psi h(t) {}_k^s \mathfrak{J}_{a^+, \omega}^\beta \varphi(t)
$$
$$
- {}_k^s \mathfrak{J}_{a^+, \omega}^\alpha h(t) {}_k^s \mathfrak{J}_{a^+, \omega}^\beta \varphi \psi(t) - {}_k^s \mathfrak{J}_{a^+, \omega}^\alpha \varphi \psi(t) {}_k^s \mathfrak{J}_{a^+, \omega}^\beta h(t)
$$
$$
+ {}_k^s \mathfrak{J}_{a^+, \omega}^\alpha \varphi(t) {}_k^s \mathfrak{J}_{a^+, \omega}^\beta \psi h(t) + {}_k^s \mathfrak{J}_{a^+, \omega}^\alpha \psi(t) {}_k^s \mathfrak{J}_{a^+, \omega}^\beta \varphi h(t), \tag{17}
$$

*where $\alpha, \beta > 0$ and $\omega(\xi) \neq 0$.*

**Proof.** Since the function $\varphi$ and $\psi$ are synchronous on $[0, \infty)$, $h \geq 0$, for all $\alpha, \beta > 0$, we have

$$
(\varphi(\xi) - \varphi(y))(\psi(\xi) - \psi(y))(h(\xi) + h(y)) \geq 0.
$$

This gives

$$
\varphi(\xi)\psi(\xi)h(\xi) + \varphi(y)\psi(y)h(y)
$$
$$
\geq \quad \varphi(\xi)\psi(y)h(\xi) + \varphi(y)\psi(\xi)h(\xi) - \varphi(y)\psi(y)h(\xi)
$$
$$
- \varphi(\xi)\psi(\xi)h(y) + \varphi(\xi)\psi(y)h(y) + \varphi(y)\psi(\xi)h(y). \tag{18}
$$

Both sides of (18) are multiplied by $\frac{(s+1)^{1-\frac{\alpha}{k}} \omega^{-1}(\xi)}{k \Gamma_k(\alpha)} (t^{s+1} - \xi^{s+1})^{\frac{\beta}{k} - 1} \omega(\xi) \xi^s$ and integrating w.r.t $\xi$ over (a,t), we obtain

$$
\frac{(s+1)^{1-\frac{\alpha}{k}} \omega^{-1}(t)}{k \Gamma_k(\alpha)} \int_a^t (t^{s+1} - \xi^{s+1})^{\frac{\alpha}{k} - 1} \xi^s \omega(\xi) \varphi(\xi) \psi(\xi) h(\xi) d\xi
$$
$$
+ \varphi(y)\psi(y)h(y) \frac{(s+1)^{1-\frac{\alpha}{k}} \omega^{-1}(t)}{k \Gamma_k(\alpha)} \int_a^t (t^{s+1} - \xi^{s+1})^{\frac{\alpha}{k} - 1} \xi^s \omega(\xi) d\xi
$$
$$
\geq \psi(y) \frac{(s+1)^{1-\frac{\alpha}{k}} \omega^{-1}(t)}{k \Gamma_k(\alpha)} \int_a^t (t^{s+1} - \xi^{s+1})^{\frac{\alpha}{k} - 1} \xi^s \omega(x) \varphi(x) h(x) dx +
$$
$$
\varphi(y) \frac{(s+1)^{1-\frac{\alpha}{k}} \omega^{-1}(t)}{k \Gamma_k(\alpha)} \int_a^t (t^{s+1} - \xi^{s+1})^{\frac{\alpha}{k} - 1} \xi^s \omega(\xi) \psi(\xi) h(\xi) d\xi
$$
$$
- \varphi(y)\psi(y) \frac{(s+1)^{1-\frac{\alpha}{k}} \omega^{-1}(t)}{k \Gamma_k(\alpha)} \int_a^t (t^{s+1} - \xi^{s+1})^{\frac{\alpha}{k} - 1} \xi^s \omega(\xi) h(\xi) d\xi
$$
$$
- h(y) \frac{(s+1)^{1-\frac{\alpha}{k}} \omega^{-1}(t)}{k \Gamma_k(\alpha)} \int_a^t (t^{s+1} - \xi^{s+1})^{\frac{\alpha}{k} - 1} \xi^s \omega(\xi) \psi(\xi) f(\xi) d\xi
$$
$$
+ \psi(y)h(y) \frac{(s+1)^{1-\frac{\alpha}{k}} \omega^{-1}(t)}{k \Gamma_k(\alpha)} \int_a^t (t^{s+1} - \xi^{s+1})^{\frac{\alpha}{k} - 1} \xi^s \omega(\xi) \varphi(\xi) d\xi
$$
$$
+ \varphi(y)h(y) \frac{(s+1)^{1-\frac{\alpha}{k}} \omega^{-1}(t)}{k \Gamma_k(\alpha)} \int_a^t (t^{s+1} - \xi^{s+1})^{\frac{\alpha}{k} - 1} \xi^s \omega(\xi) \psi(\xi) d\xi \tag{19}
$$

After multiplying both sides of (19) by $\frac{(s+1)^{1-\frac{\beta}{k}}\omega^{-1}(t)}{k\Gamma_k(\beta)}(t^{s+1}-y^{s+1})^{\frac{\beta}{k}-1}\omega(y)y^s$ and integrating w.r.t $y$ over $(a,t)$, we obtain

$$
{}^s_k\mathfrak{J}^\alpha_{a+,\omega}\varphi\psi h(t){}^s_k\mathfrak{J}^\beta_{a+,\omega}[\omega^{-1}(\xi)(1)]+{}^s_k\mathfrak{J}^\alpha_{a+,\omega}[\omega^{-1}(\xi)(1)]{}^s_k\mathfrak{J}^\beta_{a+,\omega}\varphi\psi h(t)
$$
$$
\geq{}^s_k\mathfrak{J}^\alpha_{a+,\omega}\varphi h(t){}^s_k\mathfrak{J}^\beta_{a+,\omega}\psi(t)+{}^s_k\mathfrak{J}^\alpha_{a+,\omega}g\psi h(t){}^s_k\mathfrak{J}^\beta_{a+,\omega}\varphi(t)-{}^s_k\mathfrak{J}^\alpha_{a+,\omega}h(t){}^s_k\mathfrak{J}^\beta_{a+,\omega}\varphi\psi(t)
$$
$$
-{}^s_k\mathfrak{J}^\alpha_{a+,\omega}\varphi\psi(t){}^s_k\mathfrak{J}^\beta_{a+,\omega}h(t)+{}^s_k\mathfrak{J}^\alpha_{a+,\omega}\varphi(t){}^s_k\mathfrak{J}^\beta_{a+,\omega}\psi h(t)+{}^s_k\mathfrak{J}^\alpha_{a+,\omega}\psi(t){}^s_k\mathfrak{J}^\beta_{a+,\omega}\varphi h(t),
$$

which implies

$$
\frac{\omega^{-1}(\xi)}{(s+1)^{1-\frac{\beta}{k}}\Gamma_k(\beta+k)}(t^{s+1}-a^{s+1})^{\frac{\beta}{k}-2}{}^s_k\mathfrak{J}^\alpha_{a+,\omega}\varphi\psi h(t)
$$
$$
+\frac{\omega^{-1}(\xi)}{(s+1)^{1-\frac{\alpha}{k}}\Gamma_k(\alpha+k)}(t^{s+1}-a^{s+1})^{\frac{\alpha}{k}-2}{}^s_k\mathfrak{J}^\beta_{a+,\omega}\varphi\psi h(t)
$$
$$
\geq{}^s_k\mathfrak{J}^\alpha_{a+,\omega}\varphi h(t){}^s_k\mathfrak{J}^\beta_{a+,\omega}\psi(t)+{}^s_k\mathfrak{J}^\alpha_{a+,\omega}\psi h(t){}^s_k\mathfrak{J}^\beta_{a+,\omega}\varphi(t)-{}^s_k\mathfrak{J}^\alpha_{a+,\omega}h(t){}^s_k\mathfrak{J}^\beta_{a+,\omega}\varphi\psi(t)
$$
$$
-{}^s_k\mathfrak{J}^\alpha_{a+,\omega}\varphi\psi(t){}^s_k\mathfrak{J}^\beta_{a+,\omega}h(t)+{}^s_k\mathfrak{J}^\alpha_{a+,\omega}\varphi(t){}^s_k\mathfrak{J}^\beta_{a+,\omega}\psi h(t)+{}^s_k\mathfrak{J}^\alpha_{a+,\omega}\psi(t){}^s_k\mathfrak{J}^\beta_{a+,\omega}\varphi h(t).
$$

Hence, the result is proved. □

**Corollary 6.** *Let $\varphi$ and $\psi$ be two synchronous functions on $[0,\infty]$ and $h \geq 0$. Then for all $t > a \geq 0$, the following inequality holds:*

$$
\frac{\omega^{-1}(\xi)}{(s+1)^{1-\frac{\alpha}{k}}\Gamma_k(\alpha+k)}(t^{s+1}-a^{s+1})^{\frac{\alpha}{k}-2}{}^s_k\mathfrak{J}^\alpha_{a+,\omega}\varphi\psi h(t)
$$
$$
\geq{}^s_k\mathfrak{J}^\alpha_{a+,\omega}\varphi h(t){}^s_k\mathfrak{J}^\alpha_{a+,\omega}\psi(t)+{}^s_k\mathfrak{J}^\alpha_{a+,\omega}\psi h(t){}^s_k\mathfrak{J}^\alpha_{a+,\omega}\varphi(t)-{}^s_k\mathfrak{J}^\alpha_{a+,\omega}h(t){}^s_k\mathfrak{J}^\alpha_{a+,\omega}\varphi\psi(t), \quad (20)
$$

*where $\alpha,\beta > 0$ and $\omega(\xi) \neq 0$.*

**Proof.** If we replace $\beta$ to $\alpha$ in Theorem 7, we obtain the result (20). □

**Theorem 8.** *Let $\varphi$ $\psi$ and $h$ be three monotonic functions defined on $[0,\infty]$ and satisfying the following*

$$
(\varphi(\xi)-\varphi(y))(\psi(\xi)-\psi(y))(h(\xi)-h(y)) \geq 0.
$$

*Then for all $t > a \geq 0$, the following inequality holds:*

$$
\frac{\omega^{-1}(\xi)}{(s+1)^{1-\frac{\beta}{k}}\Gamma_k(\beta+k)}(t^{s+1}-a^{s+1})^{\frac{\beta}{k}-2}{}^s_k\mathfrak{J}^\alpha_{a+,\omega}\varphi\psi h(t)
$$
$$
-\frac{\omega^{-1}(\xi)}{(s+1)^{1-\frac{\alpha}{k}}\Gamma_k(\alpha+k)}(t^{s+1}-a^{s+1})^{\frac{\alpha}{k}-2}{}^s_k\mathfrak{J}^\beta_{a+,\omega}\varphi\psi h(t)
$$
$$
\geq{}^s_k\mathfrak{J}^\alpha_{a+,\omega}\varphi h(t){}^s_k\mathfrak{J}^\beta_{a+,\omega}\psi(t)+{}^s_k\mathfrak{J}^\alpha_{a+,\omega}\psi h(t){}^s_k\mathfrak{J}^\beta_{a+,\omega}\varphi(t)-{}^s_k\mathfrak{J}^\alpha_{a+,\omega}h(t){}^s_k\mathfrak{J}^\beta_{a+,\omega}\varphi\psi(t)
$$
$$
+{}^s_k\mathfrak{J}^\alpha_{a+,\omega}\varphi\psi(t){}^s_k\mathfrak{J}^\beta_{a+,\omega}h(t)-{}^s_k\mathfrak{J}^\alpha_{a+,\omega}\varphi(t){}^s_k\mathfrak{J}^\beta_{a+,\omega}\psi h(t)-{}^s_k\mathfrak{J}^\alpha_{a+,\omega}\psi(t){}^s_k\mathfrak{J}^\beta_{a+,\omega}\varphi h(t).
$$

*where $\alpha,\beta > 0$ and $\omega(\xi) \neq 0$.*

**Proof.** Use the same argument as in the proof of Theorem 7. □

**Theorem 9.** *Let $\varphi$ and $\psi$ be defined on $[0, \infty]$. Then for all $t > a \geq 0 \; \omega(\xi) \neq 0$, $\alpha, \beta > 0$, the following inequalities for weighted $(k, s)$-RLFI hold:*

$$\frac{\omega^{-1}(\xi)}{(s+1)^{1-\frac{\beta}{k}} \Gamma_k(\beta + k)} (t^{s+1} - a^{s+1})^{\frac{\beta}{k} - 2} {}_k^s \mathfrak{J}_{a^+,\omega}^\alpha \varphi^2(t)$$

$$+ \frac{\omega^{-1}(\xi)}{(s+1)^{1-\frac{\alpha}{k}} \Gamma_k(\alpha + k)} (t^{s+1} - a^{s+1})^{\frac{\alpha}{k} - 2} {}_k^s \mathfrak{J}_{a^+,\omega}^\beta \psi^2(t)$$

$$\geq 2 {}_k^s \mathfrak{J}_{a^+,\omega}^\alpha \varphi(t) {}_k^s \mathfrak{J}_{a^+,\omega}^\beta \psi(t) \tag{21}$$

*and*

$${}_k^s \mathfrak{J}_{a^+,\omega}^\alpha \varphi^2(t) {}_k^s \mathfrak{J}_{a^+,\omega}^\beta \psi^2(t) + {}_k^s \mathfrak{J}_{a^+,\omega}^\beta \varphi^2(t) {}_k^s \mathfrak{J}_{a^+,\omega}^\alpha \psi^2(t) \quad \geq \quad 2 {}_k^s \mathfrak{J}_{a^+,\omega}^\alpha \varphi\psi(t) {}_k^s \mathfrak{J}_{a^+,\omega}^\beta \varphi\psi(t) \tag{22}$$

**Proof.** Since $(\varphi(\xi) - \psi(y))^2 \geq 0$ and $(\varphi(\xi)\psi(y) - \varphi(y)\psi(\xi))^2 \geq 0$ using the same argument as the proof in Theorem 7, we obtain (22) and (21). □

**Corollary 7.** *We have*

$$\frac{\omega^{-1}(\xi)}{(s+1)^{1-\frac{\alpha}{k}} \Gamma_k(\alpha + k)} (t^{s+1} - a^{s+1})^{\frac{\alpha}{k} - 2} [{}_k^s \mathfrak{J}_{a^+,\omega}^\alpha \varphi^2(t) + {}_k^s \mathfrak{J}_{a^+,\omega}^\beta \psi^2(t)]$$

$$\geq 2 {}_k^s \mathfrak{J}_{a^+,\omega}^\alpha \varphi(t) {}_k^s \mathfrak{J}_{a^+,\omega}^\beta \psi(t) \tag{23}$$

*and*

$${}_k^s \mathfrak{J}_{a^+,\omega}^\alpha \varphi^2(t) {}_k^s \mathfrak{J}_{a^+,\omega}^\alpha \psi^2(t) \quad \geq \quad [{}_k^s \mathfrak{J}_{a^+,\omega}^\alpha \varphi\psi(t)]^2. \tag{24}$$

**Proof.** If we replace $\beta$ to $\alpha$ in Theorem 9, we obtain the inequalities (23) and (24). □

**Remark 5.** *If we set $\omega(\xi) = 1$ in Theorems 6–9, then we obtain the inequalities of Theorems 3.1, 3.2, 3.4, and 3.5, respectively, given in [21].*

**Theorem 10.** *Let $\varphi : \mathbb{R} \to \mathbb{R}$ with $\overline{\varphi}(\xi) := \int_a^t \omega(t) t^s \varphi(t) dt$, for all $\xi > a \geq 0$, $s \in \mathbb{R} \backslash \{-1\}$. Then $\alpha \geq k > 0$ and $\omega(\xi) \neq 0$, we have*

$${}_k^s \mathfrak{J}_{a^+,\omega}^{\alpha+k} \varphi(\xi) = \frac{1}{k} {}_k^s \mathfrak{J}_{a^+,\omega}^\alpha [\omega^{-1}(\xi) \overline{\varphi}(\xi)]. \tag{25}$$

**Proof.** By using Definition 7 and the Dirichlet's formula, we have

$${}_k^s \mathfrak{J}_{a^+,\omega}^\alpha [\omega^{-1}(\xi) \overline{\varphi}(\xi)]$$

$$= \frac{(s+1)^{1-\frac{\alpha}{k}} \omega^{-1}(\xi)}{k \Gamma_k(\alpha)} \int_a^\xi (\xi^{s+1} - t^{s+1})^{\frac{\alpha}{k} - 1} t^s \omega^{-1}(t) \omega(t) \overline{\varphi}(t) dt$$

$$= \frac{(s+1)^{1-\frac{\alpha}{k}} \omega^{-1}(\xi)}{k \Gamma_k(\alpha)} \int_a^\xi (\xi^{s+1} - t^{s+1})^{\frac{\alpha}{k} - 1} t^s [\int_a^t u^s \varphi(u) \omega(u) du] dt$$

$$= \frac{(s+1)^{1-\frac{\alpha}{k}} \omega^{-1}(\xi)}{k \Gamma_k(\alpha)} \int_a^\xi u^s \varphi(u) \omega(u) [\int_u^\xi (\xi^{s+1} - t^{s+1})^{\frac{\alpha}{k} - 1} t^s dt] du$$

$$= \frac{(s+1)^{-\frac{\alpha}{k}} \omega^{-1}(\xi)}{\Gamma_k(\alpha + k)} \int_a^\xi (\xi^{s+1} - u^{s+1})^{\frac{\alpha}{k}} u^s \omega(u) \varphi(u) du$$

$$= k {}_k^s \mathfrak{J}_{a^+,\omega}^{\alpha+k} \varphi(\xi).$$

Hence, we obtained the desired result. □

## 4. The Weighted Laplace Transform of the Weighted Fractional Operators

In this section, we apply the weighted laplace transformation to the new fractional operators. For this purpose we need to substitute $\psi(t) = t^{s+1}$ on the right side of (3), where we have

$$\mathcal{L}_\psi^\omega\{\varphi(t)\}(u) = (s+1)\int_a^\infty e^{-u(t^{s+1}-a^{s+1})}\omega(t)t^s\varphi(t)dt, \tag{26}$$

which holds for all values of $u$.

**Proposition 1.**

$$\mathcal{L}_\psi^\omega\{\omega^{-1}(\xi)(\xi^{s+1}-a^{s+1})^{\frac{\alpha}{k}-1}\}(u) = \frac{\Gamma(\frac{\alpha}{k})}{u^{\frac{\alpha}{k}}}, \quad u > 0.$$

**Proof.** By using (26), we have

$$\mathcal{L}_\psi^\omega\{\omega^{-1}(\xi)(\xi^{s+1}-a^{s+1})^{\frac{\alpha}{k}-1}\}(u)$$
$$= (s+1)\int_a^\infty e^{-u(\xi^{s+1}-a^{s+1})}(\xi^{s+1}-a^{s+1})^{\frac{\alpha}{k}-1}\xi^s d\xi. \tag{27}$$

By substituting $t = (\xi^{s+1}-a^{s+1})$ on the right side of (27), we obtain

$$\mathcal{L}_\psi^\omega\{\omega^{-1}(\xi)(\xi^{s+1}-a^{s+1})^{\frac{\alpha}{k}-1}\}(u)$$
$$= \int_0^\infty e^{-ut}t^{\frac{\alpha}{k}-1}dt$$
$$= \int_0^\infty e^{-ut}\frac{(ut)^{\frac{\alpha}{k}-1}}{(u)^{\frac{\alpha}{k}-1}}\frac{u}{u}dt$$
$$= \frac{1}{u^{\frac{\alpha}{k}}}\int_0^\infty e^{-ut}(ut)^{\frac{\alpha}{k}-1}udt,$$

which gives the required result. $\square$

**Theorem 11.** *Let $\varphi$ be a piecewise continuous function on each interval $[a, \xi]$ and of weighted $\psi$-exponential order. Then*

$$\mathcal{L}_\psi^\omega\{(^s_k\mathfrak{J}^\alpha_{a^+,\omega}\varphi)(\xi)\}(u) = ((s+1)uk)^{\frac{-\alpha}{k}}\mathcal{L}_\psi^\omega\{\varphi(\xi)\}(u),$$

*where $k > 0$, $\omega(\xi) \neq 0$, $s \in \mathbb{R}\backslash\{-1\}$.*

**Proof.** By using Definitions 6 and 7 and Proposition 1, we have

$$\mathcal{L}_\psi^\omega\{(^s_k\mathfrak{J}^\alpha_{a^+,\omega}\varphi)(\xi)\}(u)$$
$$= \mathcal{L}_\psi^\omega\left\{\frac{(s+1)^{1-\frac{\alpha}{k}}\omega^{-1}(\xi)}{k\Gamma_k(\alpha)}\int_a^\xi(\xi^{s+1}-t^{s+1})^{\frac{\alpha}{k}-1}t^s\omega(t)\varphi(t)dt\right\}(u)$$
$$= \frac{(s+1)^{-\frac{\alpha}{k}}}{k\Gamma_k(\alpha)}\mathcal{L}_\psi^\omega\{\omega^{-1}(\xi)(\xi^{s+1}-t^{s+1})^{\frac{\alpha}{k}-1}*_\psi^\omega\varphi(\xi)\}(u)$$
$$= \frac{(s+1)^{-\frac{\alpha}{k}}}{k\Gamma_k(\alpha)}\mathcal{L}_\psi^\omega\{\omega^{-1}(\xi)(\xi^{s+1}-t^{s+1})^{\frac{\alpha}{k}-1}\}(u)\mathcal{L}_\psi^\omega\{\varphi(\xi)\}(u)$$
$$= \frac{(s+1)^{-\frac{\alpha}{k}}}{k\Gamma_k(\alpha)}\frac{\Gamma(\frac{\alpha}{k})}{u^{\frac{\alpha}{k}}}\mathcal{L}_\psi^\omega\{\varphi(\xi)\}(u)$$
$$= ((s+1)uk)^{-\frac{\alpha}{k}}\mathcal{L}_\psi^\omega\{\varphi(\xi)\}(u). \tag{28}$$

This proves the claimed result. $\square$

**Theorem 12.** *The Laplace transform of the weighted $(k,s)$-Riemann Liouville derivative is given by*

$$\mathfrak{L}_{\psi}^{\omega}\{(_k^s\mathfrak{D}_{a^+,\omega}^{\alpha}\varphi)(\xi)\}(u)$$
$$= (s+1)^{-\frac{nk-\alpha}{k}}(ku)^{\frac{\alpha}{k}}\mathfrak{L}_{\psi}^{\omega}\{\varphi(\xi)\}(u)$$
$$- k^n\sum_{m=0}^{n-1}u^{n-m-1}(_k^s\mathfrak{J}_{a^+,\omega}^{nk-\alpha}\varphi)_m(a^+). \tag{29}$$

**Proof.** By using Definition 9, Theorems 1 and 11, we obtain

$$\mathfrak{L}_{\psi}^{\omega}\{(_k^s\mathfrak{D}_{a^+,\omega}^{\alpha}\varphi)(\xi)\}(u)$$
$$= \mathfrak{L}_{\psi}^{\omega}\{(\xi^{1-s}\frac{d}{d\xi})^n k^n(_k^s\mathfrak{J}_{a^+,\omega}^{nk-\alpha}\varphi)(t)\}(u)$$
$$= k^n u^n \mathfrak{L}_{\psi}^{\omega}\{(_k^s\mathfrak{J}_{a^+,\omega}^{nk-\alpha}\varphi)(t)\}(u)$$
$$- k^n\sum_{m=0}^{n-1}u^{n-m-1}(_k^s\mathfrak{J}_{a^+,\omega}^{nk-\alpha}\varphi)_k(a^+)$$
$$= (uk)^n((s+1)uk)^{\frac{nk-\alpha}{k}}\mathfrak{L}_{\psi}^{\omega}\{\varphi(\xi)\}(u)$$
$$- k^n\sum_{m=0}^{n-1}u^{n-m-1}(_k^s\mathfrak{J}_{a^+,\omega}^{nk-\alpha}\varphi)_k(a^+)$$
$$= (s+1)^{\frac{nk-\alpha}{k}}(ku)^{\frac{\alpha}{k}}\mathfrak{L}_{\psi}^{\omega}\{\varphi(\xi)\}(u)$$
$$- k^n\sum_{m=0}^{n-1}u^{n-m-1}(_k^s\mathfrak{J}_{a^+,\omega}^{nk-\alpha}\varphi)_k(a^+),$$

which gives the required series solution. □

## 5. Fractional Kinetic Differ-Integral Equation

The fractional differential equations are significant in the field of applied science and have gained interest in dynamic systems, physics, and engineering. In the previous decade, the fractional kinetic equation has gained interest due to the discovery of its relationship with the CTRW theory [28]. The kinetic equations are essential in natural sciences and mathematical physics that explain the continuation of motion of the material. The generalized weighted fractional kinetic equation and its solution related to novel operators are discussed in this section. Consider the fractional kinetic equation given by

$$a(_k^s\mathfrak{D}_{0^+,\omega}^{\alpha}N)(t) - N_0\varphi(t) = b(_k^s\mathfrak{J}_{0^+,\omega}^{\beta}N)(t), \quad \varphi \in L^1[0,\infty), \tag{30}$$

with initial condition

$$\omega(0)(_k^s\mathfrak{J}_{0^+,\omega}^{nk-\alpha}N)(0) = d, \quad d \geq 0, \tag{31}$$

where $\alpha \geq 0$, $a, b \in R(a \neq 0)$, $k > 0$, $n = [\frac{\alpha}{k}] = 1$.

**Theorem 13.** *The solution of (30) with initial condition (31) is*

$$N(t) = d\omega^{-1}(t)\sum_{m=0}^{\infty}(\frac{a}{b})^n\frac{(s+1)^{\beta+(1+n)k}}{\Gamma_k(\alpha+(\alpha+\beta)n)}(\xi^{s+1}-a^{s+1})^{\frac{\alpha+(\alpha+\beta)n}{k}}$$
$$+ \frac{N_0}{a}\sum_{m=0}^{\infty}(s+1)^{\beta+(1+n)k}(_k^s\mathfrak{J}_{0^+,\omega}^{(\alpha+\beta)n+\alpha}\varphi)(t). \tag{32}$$

**Proof.** Applying the modified weighted Laplace transform on both side of (30), we obtain

$$a\mathfrak{L}_\psi^\omega\{(_k^s\mathfrak{D}_{0^+,\omega}^\alpha N)(t)\}(u) - \mathfrak{L}_\psi^\omega\{N_0\varphi(t)\}(u) = b\mathfrak{L}_\psi^\omega\{(_k^s\mathfrak{I}_{0^+,\omega}^\beta N)(t)\}(u).$$

Using Theorems 11 and 12, we obtain

$$a(s+1)^{-\frac{k-\alpha}{k}}(ku)^{\frac{\alpha}{k}}\mathfrak{L}_\psi^\omega\{N(t)\}(u) - kw(0)(_k^s\mathfrak{I}_{a^+,\omega}^{k-\alpha}N)(0) - N_0\mathfrak{L}_\psi^\omega\{\varphi(t)\}(u)$$
$$= b(s+1)^{\frac{-\alpha}{k}}(uk)^{\frac{-\alpha}{k}}\mathfrak{L}_\psi^\omega\{N(t)\}(u)$$

$$\left[\frac{a - b(s+1)^{-\frac{\alpha-k+\beta}{k}}(ku)^{-\frac{\alpha+\beta}{k}}}{(s+1)^{-\frac{\alpha-k}{k}}(ku)^{-\frac{\alpha}{k}}}\right]\mathfrak{L}_\psi^\omega\{N(t)\} = akd + N_0\mathfrak{L}_\psi^\omega\{\varphi(t)\}(u)$$

$$\mathfrak{L}_\psi^\omega\{N(t)\} = akd\left[\frac{(s+1)^{-\frac{\alpha-k}{k}}(ku)^{-\frac{\alpha}{k}}}{a - b(s+1)^{-\frac{\alpha-k+\beta}{k}}(ku)^{-\frac{\alpha+\beta}{k}}}\right]$$
$$+ \left[\frac{(s+1)^{-\frac{\alpha-k}{k}}(ku)^{-\frac{\alpha}{k}}}{a - b(s+1)^{-\frac{\alpha-k+\beta}{k}}(ku)^{-\frac{\alpha+\beta}{k}}}\right]$$
$$\times N_0\mathfrak{L}_\psi^\omega\{\varphi(t)\}(u).$$

Taking $\left|\frac{b}{a}(s+1)^{-\frac{\alpha-k+\beta}{k}}(ku)^{-\frac{\alpha+\beta}{k}}\right| < 1$, we obtain

$$\mathfrak{L}_\psi^\omega\{N(t)\} = \left[kd\left[(s+1)^{-\frac{\alpha-k}{k}}(ku)^{-\frac{\alpha}{k}}\right] + a^{-1}N_0\left[(s+1)^{-\frac{\alpha-k}{k}}(ku)^{-\frac{\alpha}{k}}\right]\right]$$
$$\times \sum_{n=0}^\infty (\frac{b}{a})^n(s+1)^{-\frac{(\alpha-k+\beta)n}{k}}(ku)^{-\frac{(\alpha+\beta)n}{k}}\mathfrak{L}_\psi^\omega\{\varphi(t)\}(u)$$
$$= kd\left[(s+1)^{-\frac{\alpha-k}{k}}(ku)^{-\frac{\alpha}{k}}\right]\sum_{n=0}^\infty (\frac{b}{a})^n(s+1)^{-\frac{(\alpha-k+\beta)n}{k}}(ku)^{-\frac{(\alpha+\beta)n}{k}}$$
$$+ a^{-1}N_0\left[(s+1)^{-\frac{\alpha-k}{k}}(ku)^{-\frac{\alpha}{k}}\right]$$
$$\times \sum_{n=0}^\infty (\frac{b}{a})^n(s+1)^{-\frac{(\alpha-k+\beta)n}{k}}(ku)^{-\frac{(\alpha+\beta)n}{k}}\mathfrak{L}_\psi^\omega\{\varphi(t)\}(u)$$
$$= kd\sum_{n=0}^\infty (\frac{b}{a})^n(s+1)^{-\frac{(\alpha-k)(n+1)+n\beta}{k}}(ku)^{-\frac{(\alpha+\beta)n+\alpha}{k}}$$
$$+ \frac{N_0}{a}\sum_{n=0}^\infty (\frac{b}{a})^n(s+1)^{-\frac{(\alpha-k)(n+1)+n\beta}{k}}(ku)^{-\frac{(\alpha+\beta)n+\alpha}{k}}\mathfrak{L}_\psi^\omega\{\varphi(t)\}(u)$$
$$= kd\sum_{n=0}^\infty (\frac{b}{a})^n(s+1)^{-\frac{(\alpha-k)(n+1)+n\beta}{k}}(ku)^{-\frac{(\alpha+\beta)n+\alpha}{k}}$$
$$+ \frac{N_0}{a}\sum_{n=0}^\infty (\frac{b}{a})^n(s+1)^{-\frac{(\alpha+\beta)n+\alpha}{k}}(s+1)^{(n+1)}(ku)^{-\frac{(\alpha+\beta)n+\alpha}{k}}\mathfrak{L}_\psi^\omega\{\varphi(t)\}(u).$$

Applying inverse Laplace transform, we obtain

$$N(t) = dw^{-1}(t)\sum_{n=0}^\infty (\frac{b}{a})^n\frac{(s+1)^{-\frac{(\alpha-k)(n+1)+n\beta}{k}}}{\Gamma_k((\alpha+\beta)n+\alpha)}(\xi^{s+1} - a^{s+1})^{\frac{(\alpha+\beta)n+\alpha}{k}-1}$$
$$+ \frac{N_0}{a}\sum_{n=0}^\infty (\frac{b}{a})^n(s+1)^{(n+1)}(_k^s\mathfrak{I}_{0^+,\omega}^{(\alpha+\beta)n+\alpha}\varphi)(t).$$

The proof of the result is completed. $\square$

## 6. Conclusions and Discussion

Fractional calculus is currently one of the most widely debated topics. In the present article, we introduced the weighted versions of the $(k, s)$-RLF operators. We then investigated and examined their properties and found the weighted Laplace transform of the new operators. Significantly, these operators reduce to notable fractional operators in the literature. Other fractional operators, such as the Riemann-Liouville fractional operators and Hadamard fractional operators, show up as special cases of these weighted fractional operators with specific choices of weighted functions and operator functions. We have developed the Chebyshev inequalities by involving the introduced fractional integral operator. We developed a fractional kinetic equation and the weighted Laplace transform used to find the solution of the said model. The presented results motivate scientists to stimulate more work in such directions.

**Author Contributions:** Conceptualization, M.S.; Formal analysis, M.U.; Funding acquisition, T.A.; Investigation, S.I.; Methodology, A.K.; Project administration, T.A.; Supervision, M.S.; Writing—review and editing, N.M., All authors jointly worked on the results and they read and approved the final manuscript.

**Funding:** There is no funding available for this work.

**Institutional Review Board Statement:** Not applicable.

**Informed Consent Statement:** Not applicable.

**Data Availability Statement:** No data were used to support this study.

**Acknowledgments:** The authors Nabil Mlaiki and Thabet Abdeljawad would like to thank Prince Sultan University (PSU) for the support through the TAS research lab. The authors would like to thank for paying the article processing charges.

**Conflicts of Interest:** The authors declare no conflict of interest.

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
