# Peer review of "On Weighted (k, s)-Riemann-Liouville Fractional Operators and Solution of Fractional Kinetic Equation"

_fractalfract, doi:10.3390/fractalfract5030118_

Round 1
Reviewer 1 Report
The paper can be published.
Reviewer 2 Report
The revised version is better. I suggest to accept it.
Reviewer 3 Report
The present version of the paper is acceptable.
This manuscript is a resubmission of an earlier submission. The following is a list of the peer review reports and author responses from that submission.
Round 1
Reviewer 1 Report
In Line 23, RLFI appears first time after the abstract. Spell it out completely and put the acronym it in parentheses.
Line 31, cover should be covers
On page 5, after the fourth line, start with some text, like Thus or Further
After line 50, when starting with Proof, start with some text, like Observe that
Page 6, line 1, replace substitute with By substituting
The end of proof, the square box, move it to the right side.
After equation (2.6), replace substitute with By substituting
Page 8, line 6, replace substitute with By substituting
Line 61, The proof is done, replace with This completes the proof.
Corollary 2.15, first line Add text Then we have
Line 62, 63, remove will
before equation (3.1), after hold, put :
Line 67, add and
Line 72, 73 start with some text
Line 80, after hold put :
Line 86, write, After both sides of (3.8) are multiplied by
Line 91, after hold put :
Line 95, after hold put :
Line 99, after hold put :
Line 100, add and
Line 101, remove then
Corollary 3.6 Start with We have
Line 105, remove will, mention which inequalities in [21]
After 4.2, replace substitute with By substituting
Theorem 4.3 after is, add, given by
After line 116, after science and before have, add, and
Start (5.1) with some text, like, We consider..
Line 117, Where should be where
After (5.3), both sides of (5.1)
Replace we get everywhere to we obtain
124, its solution was evaluated? Replace evaluated with a better word
Author Response
Editor
Fractal and Fractional
We are very thankful to the respected reviewers for positive recommendations of our article in prestigious journal.
First reviewer’s comments
The reviewer required minor language and spell check and mentioned typo’s in the article which are helpful to improve the quality of article. The reviewer acknowledge all other parameters are correct.
Response to reviewer #01.
We are very thankful of the worthy reviewer on encouraging comments on our article. All the suggested typo’s are corrected and highlighted in the pdf file of the paper.
Reviewer 2 Report
In this paper, the authors introduce the weighted version of (k,s)-Riemann-Liouville fractional integral and derivative operators. The applications of the defined operators in construction of Chebysheve inequalities and fractional kinetic equation are interesting for the reader. The paper is well written and novel. The proofs were handled tediously and mathematically correct.
I recommend the paper for publication provided that the authors after considering following comments and suggestions:
- Observe the punctuation at the end of equations. See for example in (1.1) place comma at the end.
- In general, throughout the paper, start proof of the results with some text like “By considering” or “observe that” etc.
- I suggest to indicated the advantage and weakness of the method.
- See reference 1, 2, 10, 15 and 27 there are some typo’s. The reference style should be same.
- The end proof boxes need to be at the front of end line.
Author Response
Second reviewer’s comments
The second reviewer’s recommend the article by mentioning that all parameters are correct except some improvements in the introduction and presentation of the paper are required.
Response to reviewer #02
We are thankful to the reviewer on mentioning that a few changes required in the introduction. We tried our best to improve the introduction and hopefully the reviewer will be satisfied with slight changes in the introduction. Moreover, the conclusion is more improved now and the language of the paper is also improved throughout the paper. The punctuation checked and corrected throughout the paper.
Reviewer 3 Report
In this paper, the authors consider new variants of fractional operators, namely the weighted (k,s)-Riemann-Liouville fractional integral and differential operators. The first question that we have to ask is: why? The authors do not provide any justification. While reading the paper I found that the main reason was: to write a new paper. Problems analyzed in the paper, and the analysis thereof, appear as being slight variants of (a number of) former articles on the topic. There is no real breakthrough nor the new mathematical technique of analysis. The results (and proofs) are easy modifications of known facts for other types of fractional operators.
Author Response
Third reviewer’s comments
The third reviewer raised a few questions on the results of the paper and required to presents the results clearly.
Response to reviewer #02
We are also thankful to the third reviewer on raising a few valid questions on our results.
The answer to first question that why we are studied this weighted extension is very clear that in fractional calculus the extensions in weighted sense is not an old theory. The first article was published last year in 2020 Q1 ranked journal with a title “ON THE WEIGHTED FRACTIONAL OPERATORS OF A FUNCTION WITH RESPECT TO ANOTHER FUNCTION” by Fahad Jarad and after that there are many other papers were presented in several reputed journals with weighted forms. Moreover, if we talk about weighted version in mathematics then is also an old theory for example in inequalities many classical inequalities have weighted forms like “Opial inequality.”
Secondly, reviewer said that paper contains easy modifications. The answer to that question the modifications are not as easy as we thought. There are still many classical operators that are not yet convertible to weighted forms due to their structure. I finally like to mention to the respected reviewer that our results are more general as we can obtained several classical as well as extended form with our results that are already published in Q1 and Q2 ranked journals.
Round 2
Reviewer 3 Report
Unfortunately, the answers to my questions are rather far from what I expected. The motivation of the authors for this research is that: other authors publish similar facts for another type of fractional derivatives in Q1 and Q2 ranked journals.
Why do we need a new type of derivative? Is a fractional kinetic differ-integral equation (5.1)-(5.2) describe the real model better? If yes, then please provide justification.
Next, what are the difficulties in the proof of results when compared with proofs of similar facts for another type of fractional derivative? This should be emphasized by remarks in the manuscript.
Author Response
The structure of the kernel reflecting the memory effect is crucial in describing any natural or physical phenomena. It also reflects the type of analysis followed. In this article we used a fractional derivative with a very general kernel involving a weight. Such a kernel allows describing the model in many possibilities. Whether the used specific derivative describes accurately the physical model needs experimental data which is not considered in the scope of this study. Moreover, the studied model by means of this general fractional derivative and its analysis generalizes the previous studied such models in literature. We have already emphasized by providing the remarks (2.2, 2.5, 2.16, 2.17 and 3.7) to prove the generalization of the results. Moreover, it s an open question till now which fractional operator is better than other. Same is the case of fractional models.